# From individuals to ancestries: Towards attributing trait variation to haplotypes

Yaoling Yang [1,2*], Daniel J. Lawson [1,2*]

**1** Institute for Statistical Science, School of Mathematics, University of Bristol, Bristol, United Kingdom, **2** MRC Integrative Epidemiology Unit, Population Health Sciences, University of Bristol, Bristol, United Kingdom

\* dan.lawson@bristol.ac.uk (DJL); yaolingyang1998@gmail.com (YY)

**Data availability statement:** The UK Biobank data can be accessed by approved researchers through https://www.ukbiobank.ac.uk. We used the UK Biobank data under project 81499. POBI

## Abstract

Genome-wide association studies (GWAS) have revolutionized our understanding of the genetic basis of complex traits and diseases, but limitations in SNP-centric approaches to population stratification limit the resolution of fine-scale population structures. Here we consider the use of haplotypes to represent population structure, leveraging haplotype components (HCs) for an improved understanding of trait associations and adjustment for population stratification. Using data from the UK Biobank, we showed that HCs have stronger associations with a range of phenotypes than principal components (PCs) while containing more predictive power for birthplaces globally. In GWAS, HCs-correction identifies more genome-wide significant association signals for birthplace and lifestyle-related phenotypes, which are missed by PCs-corrected GWAS. Through thorough testing and simulation, we highlight challenges in performing ancestry-specific GWAS, underscoring the critical role of accurate local ancestry inference in studying admixed populations. We analyzed the haplotype structure of the UK Biobank in terms of 93 genetically-distinct populations, which enabled the computation of Ancestral Risk Scores (ARS) across 8 continental populations, providing insights into population-specific genetic risks for traits and diseases. By integrating haplotype information, this framework provides the potential to address challenges in population stratification, enhances GWAS resolution, and supports equitable health research by facilitating genetic studies in diverse populations.

## Author summary

Understanding how genetic differences contribute to traits and diseases across diverse populations is crucial for equitable health research. While traditional methods often miss fine-scale ancestry patterns, we explore how using haplotypes - long DNA segments - can better capture population structure and improve genetic analyses. By analyzing UK Biobank individuals, we show that HCs more accurately predict birthplaces and better estimate the effect of mutations on physical, mental and more importantly, social traits such as age of first birth and educational attainment. We highlight challenges in studying

was accessed using accession no. EGAS00001000672. The HapMap3 variants list can be accessed at https://ftp.ncbi.nlm.nih.gov/hapmap/. All genetic data used in constructing the reference panel is provided by third parties and is available for use by others. The genetic map build GRCh37/hg19 is available from https://bochet.gcc.biostat.washington.edu/beagle/genetic_maps/. The code for the simulation of ancestry-specific GWAS is available on GitHub at https://github.com/YaolingYang/sim_ancestry_GWAS. The pipeline of using SparsePainter to paint UK Biobank individuals and using PBWTpaint to compute HCs are available on GitHub at https://github.com/YaolingYang/SparsePainter/tree/main/painting-pipeline.

**Funding:** Y.Y. received a PhD scholarship from the China Scholarship Council [grant number 202108060092]. The funders had no role in study design, data collection and analysis, decision to publish, or preparation of the manuscript.

**Competing interests:** The authors have declared that no competing interests exist.

admixed populations, where mixed ancestry complicates genetic analyses, and demonstrate that uncertainties in estimating ancestry at specific genome regions can limit the accuracy of ancestry-specific findings. Additionally, we introduce a robust statistical test for making genetic scores of historical populations from modern admixed individuals, reflecting ancestry-specific differences that evolved in response to varying selection pressures. In summary, our work builds a deeper understanding of population structure in admixed populations and enables the refinement of genetic analyses to capture fine-scale ancestral differences, ultimately paving the way for more personalized and equitable healthcare solutions.

## Introduction

GWAS have led to a great revolution in our understanding of the genetic basis of complex traits and diseases. These studies typically aim to identify associations between genetic variants and phenotypes, and they often require careful adjustment for population structure to avoid confounding. Population structure refers to the accumulated difference between ancestries due to independent evolution, which can bias GWAS results if it is not adequately adjusted [1].

Whilst SNPs can predict trait variation - for example, 40% of observed phenotypic variance is predictable by common SNPs and up to 90% including 5M rare SNPs [2], this may not be causal and taking a SNP-centric perspective on confounding population structure has limitations. SNPs change frequency only slowly, so careful modelling work is required to detect fine-scale structures [3–5]. Historically, principal component analysis (PCA) has been the most popular method for adjusting for population structure [6,7]. By accounting for population structure [8], PCs help reduce confounding effects, therefore improving the accuracy of association estimates and minimizing false-positive findings.

In response to these limitations, other approaches have emerged that use the idea of haplotypes [3,9,10] which leverage the inheritance process in which recombination breaks up the genome in long, continuous chunks - often measured as 'identity by descent' [11]. Whilst for very recent timescales these describe family trees and pedigrees [12,13], after just 20 generations (around 500 years) we each have 1 million ancestors, who can no longer be distinct, and the sharing of ancestors into the past behaves increasingly statistically. Learning such haplotype sharing allows the estimation of fine-scale population structure [3–5].

A haplotype-centric view of population structure has many potential applications in understanding trait associations. This paper brings the discussion up to date and performs additional experiments to establish what haplotype approaches are likely to be useful, as well as discussing natural issues for which they may be less important. We will examine the informational content in the data, before considering uses in three broad categories: SNP associations via GWAS, ancestry of genomic segments, and whole-genome summaries based on ancestry-specific polygenic scores.

For correcting for population structure when learning genetic associations in GWAS, the most common approach is to correct for genome-wide genetic similarity, either measured through the Genetic Relatedness Matrix [14,15] or PCA. These approaches all leverage SNP-level genetic similarity. Haplotype-centric approaches have been recently developed, with ancestry components [16] comparing a GWAS dataset to a pre-defined set of reference populations to identify haplotypes. Haplotype-sharing patterns learned within the GWAS dataset measured by HCs [17] are directly comparable to PCs. By leveraging the computational efficiency of the 'Positional Burrows-Wheeler Transform' [18], HCs become well-suited for large

datasets such as the UK Biobank [17]. Notably, HCs predict top PCs and retain information that is not captured by PCs, such as finer-scale associations with birthplaces, as Fig 1 illustrates (discussed completely in Results).

Building on these initial findings, our first goal is to systematically compare the efficacy of HCs and PCs in representing genome-wide ancestry and controlling for ancestral confounders in GWAS and related techniques. We assess predictive power across a wide range of continuous and binary phenotypes in the UK Biobank [19], using regression models to compute the coefficient of determination ($R^2$) for each approach. In addition, expanding on the success of birthplace prediction within the UK [17], we broaden this comparison to include worldwide populations. We also perform GWAS on UK Biobank traits correcting for population stratification using each measure to identify SNPs that show significant differences in association when controlled with HCs versus PCs.

The second goal is to consider applications in admixed datasets. Whilst migration and globalization lead to more people having ancestry from multiple ancestral populations, these admixed individuals have often been excluded from large-scale genomic studies due to challenges related to population structure [20]. The complexity arises because population substructure can lead to false positives in GWAS if not effectively controlled. Conventional approaches, like using PCs to adjust for broad population structure, are ineffective for admixed populations because they only capture global ancestry proportions and not the specific local ancestry at each genomic locus [21]. As a result, the inability to handle local ancestry has limited the clinical utility of large-scale genomic datasets for admixed populations, contributing to ongoing health disparities [22–24].

Local ancestry inference (LAI) [17,25–28] offers hope for working with admixed populations. Several attempts have been made to leverage LAI to examine the trait-gene associations, such as local ancestry PC correction [29], the inclusion of local ancestries as covariates to reduce confounding [30], and a two-step local ancestry adjusted testing procedure named LAAA [31]. Methods such as AsaMap do not rely on accurate LAI [32] but do not considerably improve power compared with standard GWAS.

More recently, the Tractor framework [21] enables the inclusion of admixed individuals in GWAS by accounting for both the global and local ancestral composition of each individual, providing ancestry-specific effect size estimates. Since we can now compute LAI at scale it is natural to deploy Tractor-like models on biobanks. By simulating different levels of uncertainty in local ancestry calls, we aim to understand how these uncertainties affect the performance and accuracy of the Tractor model in estimating ancestry-specific effect sizes. These simulations are crucial for assessing the feasibility of using Tractor in practice, but unfortunately, we will show that uncertainty in LAI prevents estimating ancestry-specific effect sizes for rare SNPs in this framework. Fortunately for our understanding of the underlying biology, there is increasing evidence that the majority of traits lack ancestry-specific effects [16].

Our final goal is to use local ancestry to estimate ancestral contributions to genetic traits. The comparison of homogeneous populations with different ethnicities has revealed different genome-wide trait associations or genetic scores [33,34]. Given access to local ancestry information, it is natural to attribute trait association across ancestries in an 'ancestral risk score' [35,36]. We will show that ARS can quantify the genetic risk of each ancestry accounting for local ancestry probabilities. This flexibility makes ARS particularly suitable for studies with high uncertainty in local ancestry estimates, as it can still provide meaningful insights into ancestry-specific risks without requiring definitive ancestry calls at each locus. In this work, we painted the UK Biobank using a fine-scale reference panel with 4,334 reference individuals from public-available data sources (i.e. all except POPRES) as used in Hu et al. [16], and applied ARS to compute ancestry-specific risk scores across multiple diseases.

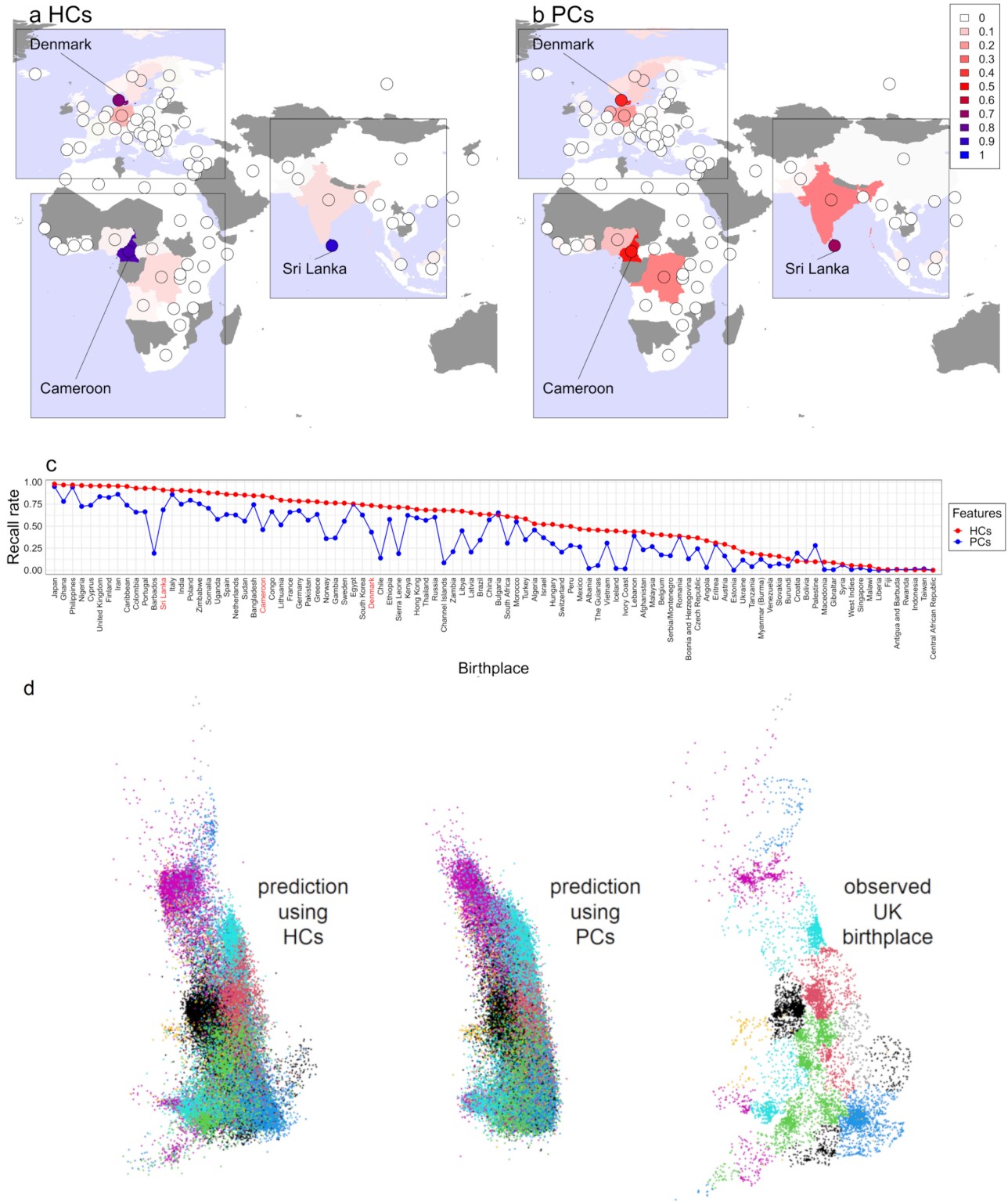

**Fig 1. Predicting birthplaces using 150 HCs or PCs in a left-out sample of UK Biobank using xgboost.** a-c, prediction of birth Country for 20,092 internationally born non-British ancestry individuals, broken down by birth country. a-b, example countries, visualized in a world map. c, the recall rate (higher is better) contrasting HCs and PCs. d, visualising information about birth location present in HCs (left) or PCs (middle) using xgboost prediction of the UK birthplace of n = 341,881 British-born individuals with self-reported British ethnicity, coloured by ancestral region of birthplace (right). The UK and world map data are available at https://gadm.org and through R package 'rworldmap' at https://cran.r-project.org/web/packages/rworldmap/index.html.

## Results

### Prediction of worldwide birthplaces using HCs and PCs

To illustrate the population structure captured by HCs and PCs, we assessed their ability to predict individuals' birthplaces using a dataset of 20,092 non-British ethnicity UK Biobank participants from 98 worldwide birthplaces (Methods). Using XGBoost models trained on the top 150 HCs or PCs, we found that models based on HCs had a notably higher average recall rate (80.19%) than those based on PCs (62.68%), indicating that HCs have stronger associations with birthplaces than PCs. The two countries for which PC predictions improve on HCs are Croatia and Palestine with 41 and 21 individuals respectively of the 20,092 total used, so this may reflect statistical noise or imperfect labels, as many individuals will trace ancestry back to places other than their location of birth. The average probabilities of correctly predicting each birthplace, i.e. the recall rate, were visualized in Fig 1c and confusion matrices in S1 Fig, which highlights the superior predictive performance of HCs in capturing fine-scale population structure. For example, PCs failed to distinguish Denmark from nearby European countries, and Cameroon from adjacent African countries, and PCs mistake a great proportion of Sri Lanka people as Indian (Fig 1b). By contrast, HCs show much higher prediction accuracy in those birthplaces (Fig 1a).

Fig 1d applies the XGBoost pipeline within the UK to visualize how HCs capture subtle population structure lost by PCs, including drift components from Eastern, South Eastern, and South Western England and North-East Scotland.

### Comparison of UKB phenotype prediction performance of population structure represented by HCs and PCs

HCs have at least two advantages over PCs. The first is conceptual - because PCs are constructed from total variance explained, many are associated with genetically localized structures rather than population structure. HCs, being built on recent relatedness patterns, are immune to this and so any number can be used in principle. To illustrate this, Fig 2 shows how every PC>18 is localized to one or a few genomic regions as summarized with autocorrelation, with e.g. PC19 tagging a single region on chromosome 6. The second is practical - as they represent more recent relationships in finer-scale, they can be expected to capture more population structure.

To assess the information content in HCs/PCs (chosen by cross-validation), we compared the performance of using up to 18 PCs (these first 18 describe genome-wide average SNP frequency variation, whilst the rest are also mixed with genomic structure, as shown in Fig 2 and ref. [8,16]) and 150 HCs (which describe within-sample recent relationships via haplotypes) to predict phenotypes in left-out data. In out-of-sample performance, HCs explain higher variance than PCs for all continuous and nearly all binary phenotypes (Fig 3C, S1–S2 Tables), for instance, 4 times for 'educational attainment', over twice for 'age at first live birth', 'alcohol usually taken with meals', and 'cin/pre-cancer cell cervix'. Other notable increases are 'breastfed as a baby', 'weight', 'BMI', 'angina', 'cholelithiasis/gall stones', 'mood swings', etc.

To test whether constraining PCs mattered, we computed the out-of-sample performance without constraint, for top HCs/PCs for every phenotype (S2–S3 Figs), with largely unchanged qualitative conclusions. No continuous phenotype could be as well explained with PCs as with HCs, and for those few binary traits where HCs are similar or marginally outperformed by PCs, the total variance explained is tiny because those traits are very rare. As British populations dominate the UKB, the significant increase of $R^2$ observed for certain HCs suggests that these components capture variations specific to the UK. By contrast, many other

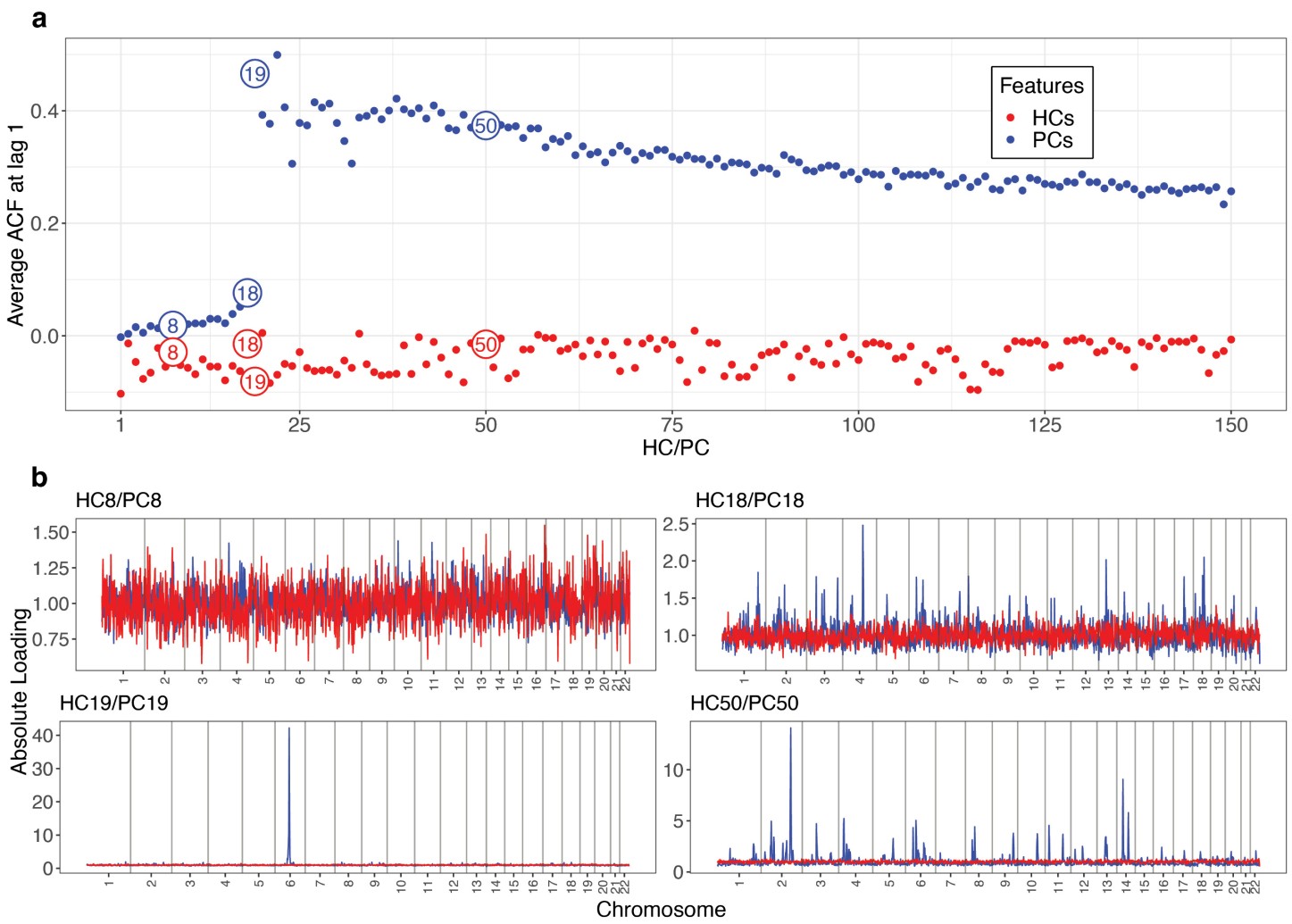

**Fig 2. LD structure as a function of HCs/PCs in the UK Biobank.** a, average auto-correlation function (ACF) at lag 1 of the absolute value of loadings for the top 150 HCs and PCs. b, aggregated absolute value of loadings from highlighted HCs/PCs (red/blue in a) throughout the genome with SNPs aggregated into bins of 100. A total of n = 406,773 individuals from the UK Biobank are included in this analysis.

HCs associated with under-represented populations in the UKB may simply add noise when predicting phenotypes.

By capturing pairwise coancestry, HCs can offer a finer-scale representation of population structure (and associated geographical or socio-economic confounding) than traditional PCs, thus leading to better variance explanation ability in GWAS and related analyses.

## GWAS with HC and PC correction

To compare the effectiveness of HCs and PCs in correcting for population structure in GWAS, we performed GWAS using the top 18 components on 26 continuous and 53 binary phenotypes, and visualized them in S4–S5 Figs.

For educational attainment, we found 6 SNPs overlapping gene ACMSD (associated with tryptophan metabolism and neurlogical disorders including parkinsonism [37]) and CCNT2-AS1 (associated with various disorders, again including Parkinson's disease [38]) which were

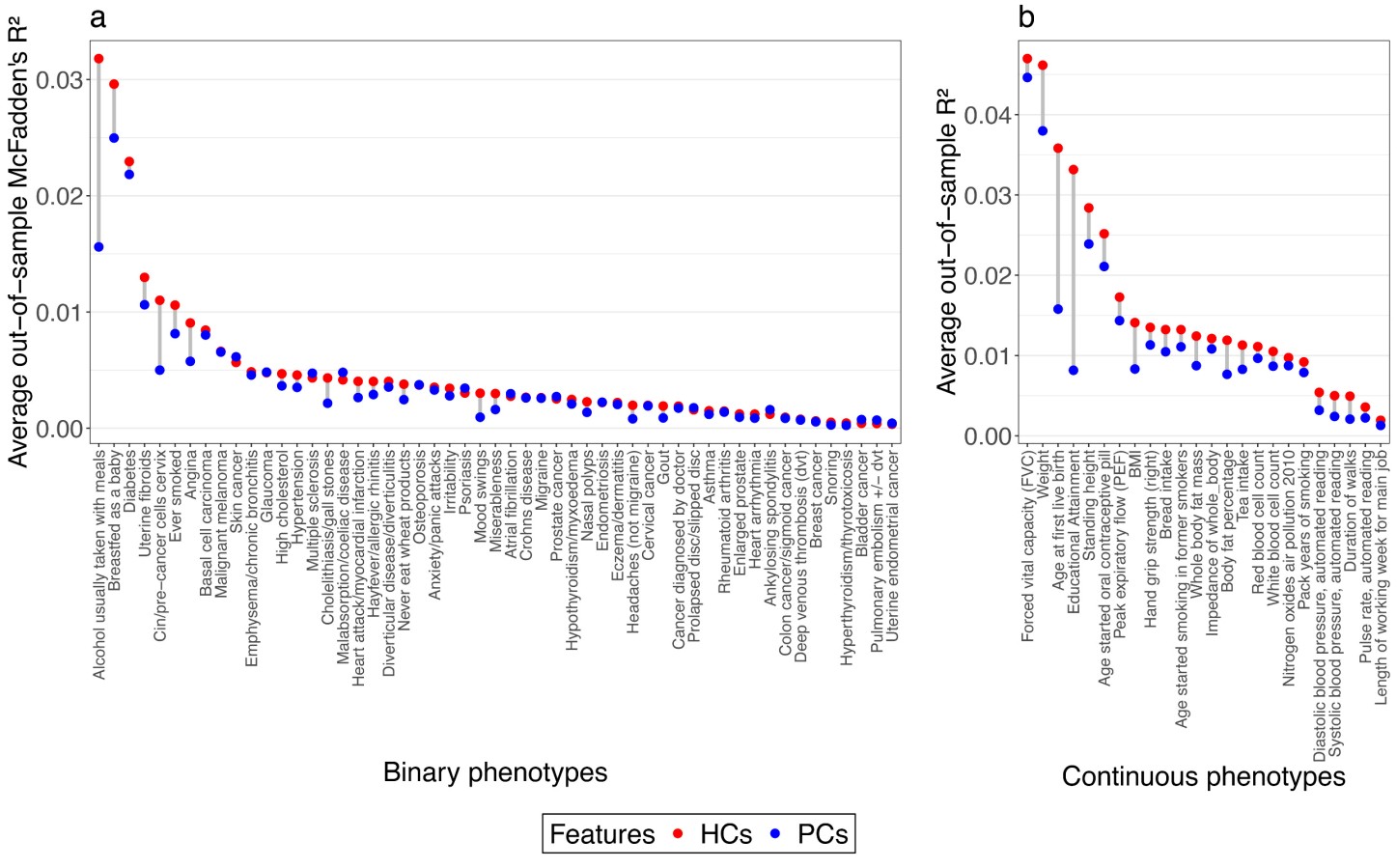

**Fig 3. Comparison of the predictive performance of HCs and PCs on phenotypes.** We visualize the average out-of-sample $R^2$ explained by HCs (red) and PCs (blue) under linear regression models (plot a, for continuous phenotypes) or logistic regression models (plot b, for binary phenotypes) through a 5-fold cross-validation performed on n = 462,694 individuals in the UK Biobank, constraining to PCs not associated with genomic structure.

significant at genome-wide significance level with HC-corrected GWAS, but had no significance ($p > 0.01$) with PC-corrected GWAS. In the UK Biobank database, these SNPs were also reported to have GWAS hits in many other phenotypes, which indicates that these might be real signals over-corrected by PCs. Because HCs are highly informative of birthplaces, we found changes in significance when correcting for HCs/PCs in both the east and north coordinates in a number of SNPs.

The p-value distribution was shifted in other phenotypes, including "Age started smoking in former smokers", "Tea intake", "Breastfed as a baby", "Alcohol usually taken with meals", "Cervical cancer" and "Cin/pre-cancer cells cervix". Whilst interesting, we don't view these as substantial differences in the UK Biobank dataset. Instead, HCs have advantages primarily in specific contexts where fine-scale population structure is important. This may include datasets that include inter-continental admixture, which is of limited scale in these data.

### Simulation reveals ancestry-specific GWAS requires accurate local ancestry assignment

We intended to estimate ancestry-specific effect sizes in the UK Biobank. To evaluate whether these results would be meaningful, we simulated local ancestry with uncertainty under the

same linear regression model used for inference in a minimal two-way admixture model. Inference failure under this model means that causal effect sizes are not separately identifiable from local ancestry effects. We compared the performance of the Tractor model [21] with different admixture fractions, MAF thresholds and LAI certainties under different methods of representing local ancestry: TRUTH uses the real LAI, RAW uses the certainty on LAI calls as a weighting, BESTGUESS treats the highest probability LAI call as a certain call, and SAMPLING randomly averages the effect size estimate from many random samples from the probability distribution (see Methods).

The Tractor framework can always return accurate estimates of effect sizes on average (Fig 4 and S6–S7 Figs) when we know the true ancestries (TRUTH), while SAMPLING and BESTGUESS are always inaccurate. By contrast, although replacing definitive ancestry calls in the Tractor framework with the local ancestry probabilities (RAW) achieves high accuracy under high certainty ($\geq 0.9$) when modelling common SNPs, i.e. MAF $\geq 20\%$, the estimates deviate considerably from the truth under imperfect LAI for less common SNPs.

This is not just a theoretical concern. Estimating LAI for the UK Biobank dataset with SparsePainter (which has comparable accuracy to other LAI estimators [17]), and using 1000 Genomes as reference, few haplotypes outside of Europe meet the threshold of 95% and 98% LAI probability (Methods, S8 Fig). Specifically, we examined the LAI from all 925k haplotypes and averaged across all pruned significant SNPs from the 35 phenotypes investigated in the ARS analysis. Retaining "certain" calls at 98% certainty we obtain a sample size of just 144 Middle East, 207 America, 452 South Asian, 750 East Asian, and 1949 African LAI segments (rising to 303, 424, 904, 1397, and 2961 respectively at 95% confidence). Uncertainty is costing a lot here - the number at only 50% confidence rises to 1.8K, 2.5K, 6.3K, 7.1K and 8.9K, respectively. The number of confidently called LAI segments is far too small for any GWAS, indicating the limited power of the UK Biobank to implement the Tractor framework in practice effectively.

Moreover, the RAW approach has substantially higher standard errors in effect size estimates S9–S11 Figs), which consequently reduces the statistical significance in practical GWAS. Given that significant GWAS signals often arise from rare SNPs, these findings indicate the challenges of accurately estimating ancestry-specific effect sizes with the Tractor framework under conditions of LAI uncertainty.

We have also explored using Bayesian modelling or variational inference to estimate the ancestry-specific effect sizes. However, with $2N(k-1)$ ancestry calls to estimate, which exceed the sample size $N$, accurately inferring the ancestry-specific effect sizes becomes almost infeasible. Real GWAS are heterogeneous and causal loci cannot be assumed to be the high-frequency tagging variants that have the most predictive power. Therefore, we conclude that while Tractor serves as an idealized model, its utility is primarily limited to contexts involving highly distinct populations. In fine-scale admixed populations, its effectiveness depends heavily on the precision of local ancestry assignments, which remains a challenge with current tools. Thus, Tractor may still be applicable in admixed populations, provided the ancestral differences are clear enough and can be accurately identified - which we will show is not true of the UK Biobank.

## ARS scan for traits across continental populations detect ancestral-specific risks

To work with quantities that are uncertainty-tolerant, we painted the UK Biobank with a public-available reference panel with 93 populations that we summarized (S3 Table), and then estimated the frequency of each SNP in each ancestry (Methods) which results in an

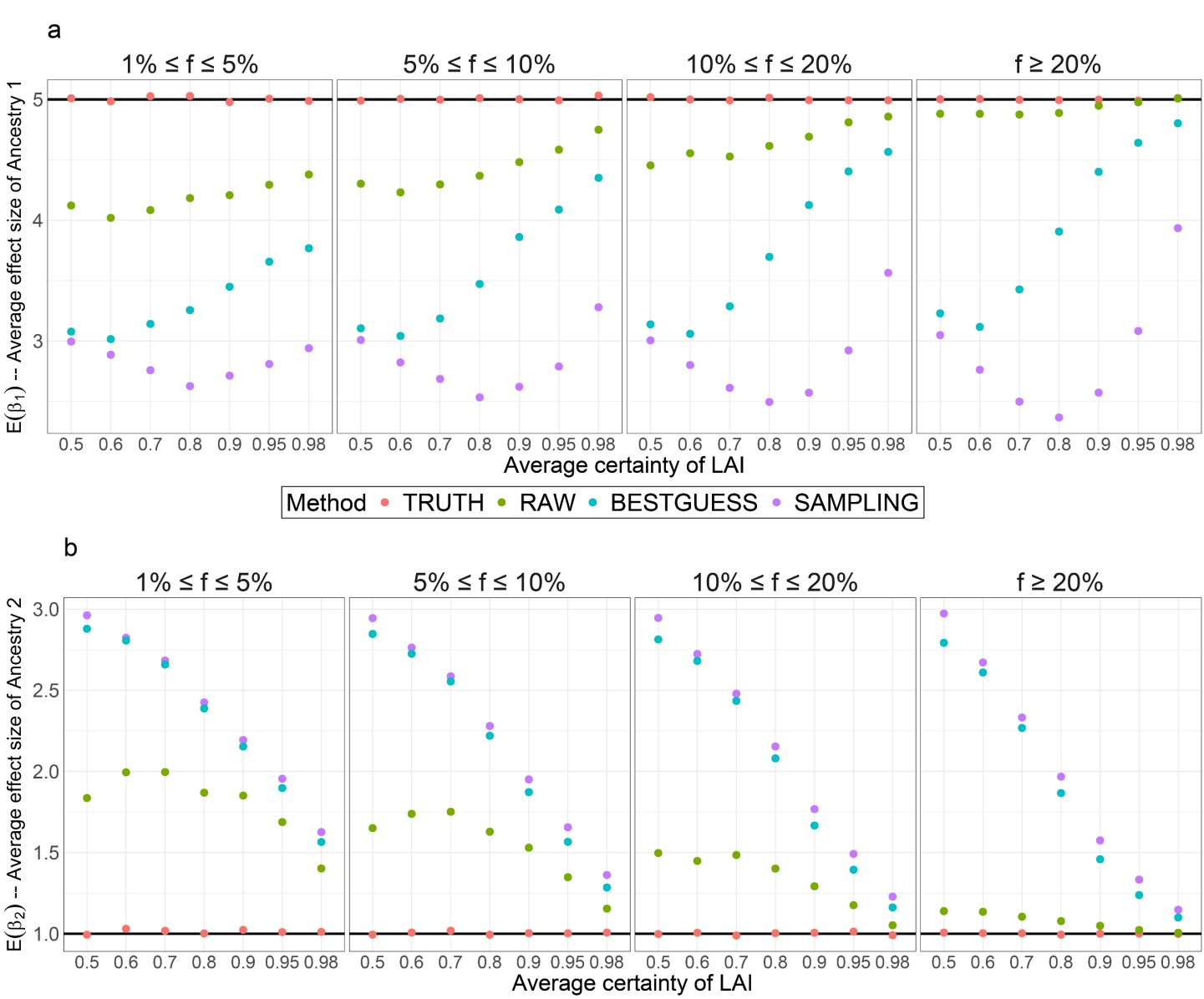

**Fig 4. Comparison of the average estimated effect size of ancestries from the Tractor model with different methods to represent local ancestries under** $\mathbf{E(A^*_{ild})} = \mathbf{0.1}$. $E(A^*_{ild})$ denotes the average probability of Ancestry 1. The x-axis represents the average certainty of LAI (see Methods) and the y-axis represents the average estimated effect sizes for Ancestry 1 (plot a) and Ancestry 2 (plot b). Different MAF thresholds $f$ are compared: $1\% \leq f \leq 5\%$, $5\% \leq f \leq 10\%$, $10\% \leq f \leq 20\%$, and $f \geq 20\%$. The simulation was repeated 1,000 times with n = 20,000 diploid individuals.

ARS when included as weights in a sum over effect sizes for a phenotype. ARS was estimated for 35 phenotypes across 8 continental populations to assess population-specific genetic risk profiles, and quantified their uncertainty.

We computed p-values using a two-sided empirical test (Methods) that accounts for variation in individuals and SNPs. We tested the null hypothesis that our associations are driven by chance sampling of ancestries for associated SNPs, both for single SNP associations (S12 Fig) or to identify ARS signals that are real outliers (Fig 5). Visualization of these results shows variability in the ARS estimate for the SNPs actually associated with the trait (confidence

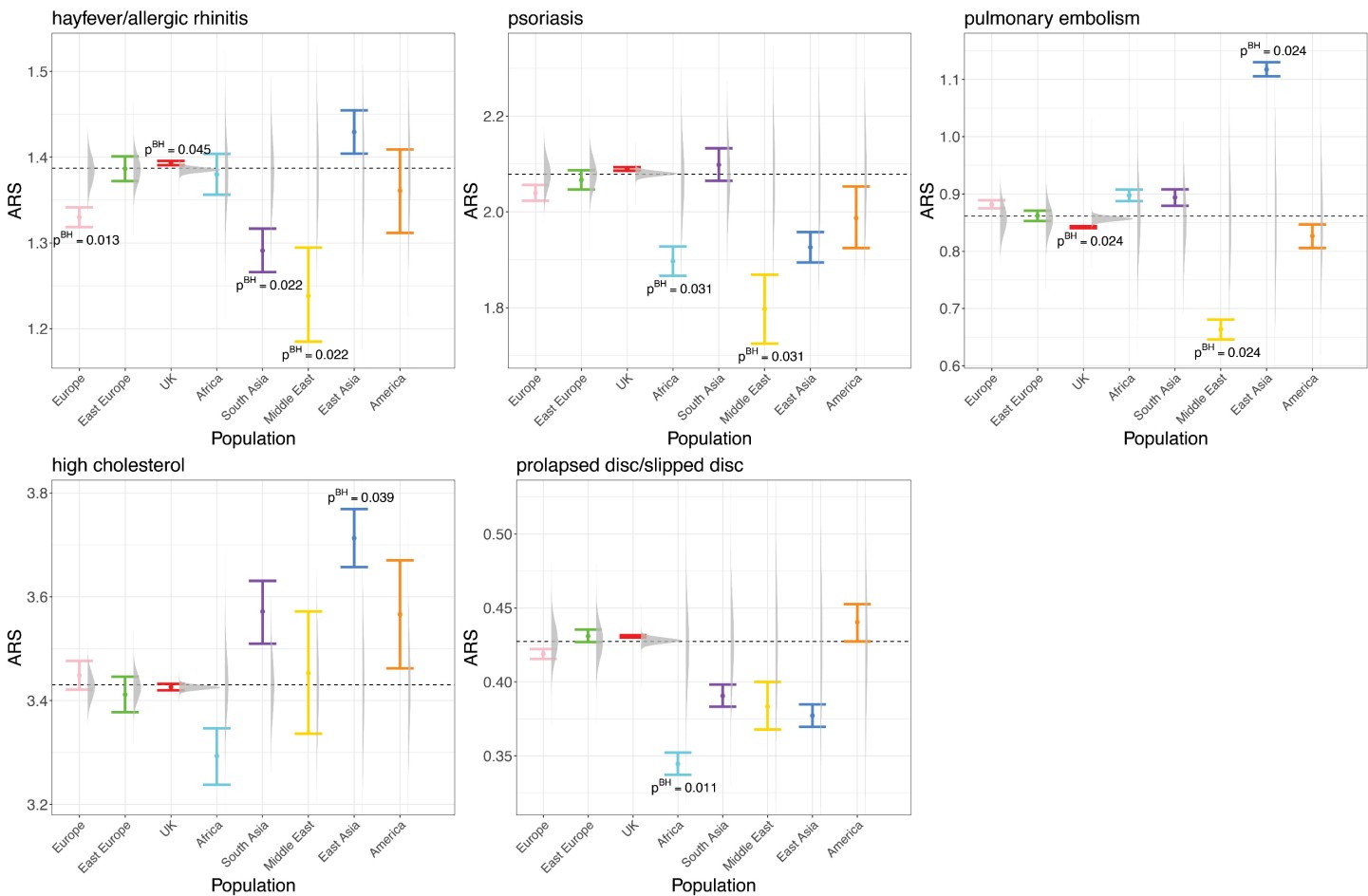

**Fig 5. ARS for 5 significant phenotypes ordered by the overall significance level.** ARS are computed from n = 462,694 individuals in the UK Biobank. The error bars represent the 95% confidence interval. The distribution of simulated ARS for each population is shown as a raincloud plot, under the null hypothesis that SNPs associated with the trait are not associated with ancestries (Methods). The dashed black lines represent the average ARS of all populations weighted by the population sizes. The ARS of populations with Benjamini-Hochberg corrected p-values significant at 5% computed from the two-sided empirical test are annotated.

intervals) as well as 'simulated ARS' that could be expected by matching SNPs with equal frequency genome-wide (grey raincloud plots). The test accounts for both of these effects together.

To assess the overall significance of each phenotype, we employed Fisher's method and adjusted the p-values using the Benjamini-Hochberg correction [39], with results visualized in S13 Fig. Fig 5 highlights the ARS for traits with significant corrected p-values (< 0.05). Due to differences in population representation, under-represented populations such as the Middle East and America show wider confidence intervals, while more represented populations, such as the UK, exhibit narrower intervals.

Overall, there are marked differences in ARS between continental ancestries, with UK separated from Europe due to increased sample size and hence has power to make this contrast. For hay fever/allergic rhinitis, UK has significantly higher risk than that found in Europe, South Asia, and the Middle East. Psoriasis is associated with reduced risks in Africa and the Middle East, aligning with World Health Organization reports indicating lower risks in Egypt and Tanzania [40]. East Asia has the highest risk for pulmonary embolism which contrasts

a significantly lower risk in the Middle East. We found significantly higher East Asia risk for high cholesterol and lower African risk for prolapsed disc/slipped disc. Other phenotypes without significant overall ARS may have indicative trends (S14 Fig).

It should be noted that these associations are found in UK residents and are based on ancestry patterns at SNPs identified in GWAS in Europeans. They therefore may not be expected to reflect observed trait associations outside of the UK. The variability in genetic risks for these phenotypes, particularly those with significant overall ARS, remains largely underexplored due to the limited availability of genetic data across global populations.

## Discussion

In this study, we summarize the current haplotype-centric framework to understand trait associations, comparing associations of traditional PCs with haplotype-based within-sample HCs in representing genome-wide ancestry and controlling for population structure in GWAS. As datasets grow in scale to include a moderate fraction of a population, such as seen in the Estonian Biobank [41] (200k of the 1.3M population), FinnGen [42] (500K of 5.6M) or Our Future Health (working towards 5M of the UKs 69M) HCs promise to bring a familial level representation of population structure. Similarly, in more diverse and admixed populations such as PAGE [43], All of Us [44] (currently 245k people from the USA) and the Million Veteran Program [45], methods that can identify local ancestry structure at scale are essential. This is true regardless of whether local ancestry is itself predictive, as explored by TRACTOR-like models [21], or just a proxy for confounding by LD as evidenced by models but without access to large diverse data [16].

We show that HCs capture finer-scale ancestral signals than those captured by PCs, thus providing a more detailed representation of genetic ancestry. HCs explain a higher proportion of variance across many phenotypes than PCs, by a factor of 2 or more for social variables such as 'educational attainment', 'age of first birth', and 'alcohol taken with meals'. When predicting birth country, HC-based models achieve a significantly higher average recall rate than those based on PCs.

In GWAS, controlling for population structure is important to avoid false positive associations. The similar performance of HCs and PCs for many traits implies that the information needed for ancestry correction is present in PCs. These results hold for relatively unadmixed and homogeneous populations like the UK Biobank, whereas HCs may be more powerful when correcting for population structure in complex admixtures between genetically separated populations.

Unlike PCs which are distributed across all populations that share any historical genetic component (e.g. [46]), HCs [17] form orthogonal patterns representing groups of individuals that share recent relatedness via genetic isolation or admixture. While this study focuses on the applications in the UK Biobank where the majority of participants are white British, applying the same methods in other biobanks with more diverse populations may show a greater advantage for HCs by separating recent relatedness in admixed groups from repeated admixture events.

Performance of estimates for effect sizes from LAI depend critically on uncertainty, leading to a need for LAI inference tools that estimate uncertainty, and leverage information - such as exploiting rare SNPs, and building models to address admixture in the panel - to reduce it. Where uncertainty can be eliminated - perhaps at the intercontinental level through careful curation of reference data and stringent thresholds - LAI-specific estimates can in principle be obtained, though very large sample sizes will be needed [47]. When SNPs have uncertain

ancestry assignment, our results caution against the application of methods such as Tractor that try to infer differing effect sizes by population. Instead, we recommend focusing on quantities that are fully specified under uncertainty. We introduced a non-parametric hypothesis test for the significance of ARS which quantifies the genetic risk of ancestries in *admixed* datasets. We showed that ARS varies for numerous diseases and traits among continental populations, which previous studies have linked to population-specific selection [36]. This highlights the importance of population-specific medical research to build a comprehensive understanding of different diseases.

Our study also has some limitations. Whilst ARS can be directly interpreted the same way that PGS are, they suffer from the same problems and potentially some new ones. Firstly, effect sizes are ascertained in Europeans and so predictive power decreases with genetic distance from this population, primarily due to LD [16]. Secondly, genome-wide distributions of effects have shifted due to European drift from the historical African population [48]. Thirdly, unlike PGS, ARS work with inferred local ancestry which may interact with LD tagging of GWAS hits. The solution in all cases is to increase the diversity of GWAS panels, and to try to construct scores from fine-mapped, putatively causal effect size estimates.

Although the reference panel we constructed from public-available datasets is comprehensive, it does not capture all the genetic diversity in global populations due to insufficient sample sizes. Improving reference datasets is critical to reducing the uncertainty of local ancestry inference, particularly for under-represented groups. Furthermore, our simulations highlighted the difficulty in estimating ancestry-specific effect sizes without precise local ancestry information. To address this, future work could develop scalable methods for local ancestry inference that account for admixed reference populations, or statistical models that can effectively use local ancestry probabilities to estimate ancestry-specific effects.

This study leads to two recommendations. Firstly, HCs improve genetic association with phenotypes compared to PCA, with application in GWAS and genome-wide prediction. Secondly, ARS is a useful and unbiased measure of genetic association with ancestry in admixed populations that use local ancestry and for which we now provide a robust association test. Conversely, we found that ancestry-specific effect sizes were underspecified in admixed populations with uncertain local ancestry calls. In conclusion, this study reviews the uses of haplotypes and local ancestry for understanding trait associations through a thorough examination of what works in the UK Biobank data, and what does not.

## Methods

### Prediction of worldwide birthplaces in the UK Biobank with HCs or PCs

We aimed to assess the ability of HCs and PCs to predict individuals' birthplaces. We summarized the number of participants in the UK Biobank from different worldwide birthplaces by their self-reported ethnic group (hereafter 'ethnicity'), i.e. British, white (excluding British), Asian, black, mixed, others or unknown. For each birthplace, we included participants from ethnicities representative of local residents only if there were at least 10 individuals for those ethnicities. Additionally, we sampled 1,000 individuals born in the UK to balance the dataset. These filtering steps resulted in a final cohort of 20,092 individuals (including both males and females) from 98 birthplaces, which were summarized in S4 Table.

We conducted an analysis to compare the prediction performance of the top 150 HCs and PCs on birthplaces with eXtreme Gradient Boosting (XGBoost [49]). First, we randomly

split the datasets with stratified sampling into an 80% training set and a 20% test set. We performed a 5-fold cross-validation (CV) on the training data to determine the optimal maximal number of boosting iterations, where 4 folds were used for training and 1 fold for validation. Then, XGBoost models were trained using 150 HCs or PCs and categorical birthplace labels in the training set, and predictions were made on the test set. This entire CV process was repeated 50 times, after which the average confusion matrix was computed for further analysis, including the visualization of the prediction results in a heatmap (S1 Fig) or a world map (Fig 1a–1b).

We repeated this analysis for the 341,881 individuals who were born in the UK and self-identified as white or British ethnicity, applying XGBoost to the Easting and Northing components to predict UK birth locations, and reported the visualization results in Fig 1d.

## Computation of loadings and autocorrelation of HCs and PCs

Let $X$ denote the $N \times L$ genotype data for $N = 406,773$ unrelated individuals (including both males and females) within the UK Biobank, as implemented by Bycroft et al. [19], and $L = 147,604$ SNPs. Let $H$ denote the $N \times K$ HC matrix for the top 150 HCs (following the computation of Yang et al. [17]). We aimed to project $H$ onto $X$ to compute the SNP loadings $V_h$. Let $X = HV_h^T$ be the decomposition of $X$ in terms of HCs. We computed $V_h = ((H^TH)^{-1}H^TX)^T$ by matrix transformation.

To compute the loadings for PCs $V_p$, we performed the singular value decomposition on $X$, i.e. $X = U_p\Sigma V_p^T$ where $\Sigma$ is a diagonal matrix of the singular values, and then we extracted $V_p$ with the top 150 components, i.e. PCs.

Next, we estimated the autocorrelation function (ACF) for both the absolute value of loadings for HCs $V_h$ (after normalization to ensure it has the same scale as $V_p$) and PCs $V_p$, and visualized the average ACF at lag 1 in Fig 2a. We also visualized the absolute value of loadings for specific HCs and PCs throughout the genome with SNPs aggregated into bins of 100 (Fig 2b).

## Comparison of prediction performance between HCs and PCs through cross-validation

To evaluate the information contained in HCs/PCs data, we compared the predictive performance of the top 150 HCs and PCs by fitting regression models on 24 continuous and 53 binary phenotypes. For each model, we used either PCs or HCs as predictors and included sex and age as covariates. Instead of simply reporting the in-sample $R^2$, we performed a 5-fold CV scheme and reported the average out-of-sample $R^2$ on the test set. In detail, we trained the model using 4 folds and tested it with 1 fold, and then we repeated the process 5 times until each fold was used as the test fold once.

We also aimed to find the optimal number of HCs/PCs with the best out-of-sample performance. Here only the top 18 PCs were considered because the remaining PCs capture complex linkage disequilibrium (LD) structure rather than solely population structure (Fig 2) [8,16]. In specific, we applied the above CV process for any of the top k HCs/PCs (k = 1,..,150 for HCs and k = 1,..,18 for PCs), and reported the optimal k with the highest average out-of-sample $R^2$.

Different measurements of $R^2$ were used for different models. We performed linear regression models on continuous phenotypes and reported the standard $R^2$; for binary phenotypes, we performed logistic regression models and used McFadden's pseudo-$R^2$ [50] to measure the

variance explained by the model. To measure the $R^2$ of only the HCs and PCs while removing the effects of sex and age, we used the subtraction of the $R^2$ of the full model (i.e. $Y \sim$ sex + age + HCs/PCs) and the $R^2$ of the model with only sex and age as covariates (i.e. $Y \sim$ sex + age).

The study was performed on 406,773 unrelated individuals (including both males and females) within the UK Biobank, as implemented by Bycroft et al. [19].

## GWAS scan with HCs or PCs correction

To compare the effectiveness of HCs and PCs in correcting for population structure in GWAS, we conducted GWAS scans on the aforementioned 24 continuous and 53 binary phenotypes, plus the east and north coordinates of the birthplace, using PLINK 2.0 [51]. The sex and age of participants are also included in the GWAS model as the standard confounding correction. The top 18 HCs and PCs are used for genome-wide population correction, respectively. Standard linear regression models are used for continuous phenotypes, and logistic regression models for binary phenotypes.

## Public-available reference panel with 93 populations

We summarized the public-available reference panel following the steps in Hu et al. [16], including HAPMAP3 [52], HGDP [53], POBI [5], Busby_Europe [54], Pagani_Africa [55], Peterson_Africa [56], Schlebusch_Africa [57], and excluding POPRES [58] (due to restricted access), which contains 4,334 individuals (including both males and females) from 129 worldwide populations. Below we performed an additional quality control on this reference panel.

We applied PBWTpaint to compute the expected length of genome matched (or 'copied'), which approximates the amount of genome the $i$th individual shares most recently with the $j$th, hereafter 'chunk length', denoted $A_{ij}$. This process requires phased genotype data. In detail, for each haplotype, we then calculated the sum of the chunk lengths weighted by the total genetic distance of each chromosome. This was performed for both haplotypes of each individual, after which we combined the two haplotypes to obtain a total measure per individual. We use $A_{ij}$ to denote the total genomic length that the $i$th individual copied from the $j$th individual.

Using the population labels of each individual provided by Hu et al. [16], we aggregated $A_{ij}$ values by population, and therefore $A$ shrinks from an $N \times N$ matrix to a $K \times K$ matrix, where $N$ is the number of individuals and $K$ is the number of populations. Next, we normalized each row by the diagonal element which refers to the expected length of copied chunks from the same population. Because we have fewer samples than Hu et al. [16], not all populations are as cleanly separated. We therefore merged populations for which the inter-individual chunk length variation overlapped iteratively until each updated population was well distinguished from others. This resulted in a refined reference panel with 93 distinct populations (S3 Table).

## Paint the UK Biobank with the public-available reference panel

We inferred the local ancestry of UK Biobank individuals using the public-available reference panel as the reference data, which includes 4,334 individuals from 93 populations. We filtered the common bi-allelic SNPs with minor allele frequency MAF $\geq$ 5% in both the imputed UK Biobank dataset and the reference dataset, resulting in a total of 667,543 SNPs. Next, these two datasets were merged and then phased using Beagle 5.4 [59]. Finally, we split the phased dataset into the reference and target datasets according to their labels.

We did an additional filtering of the UK Biobank individuals by removing individuals who have at least one of the grandparents in common. In detail, we computed the pairwise genetic

relatedness score within each chromosome using the '–genome' command with PLINK 1.9 [51], and then computed the average of them weighted by the genetic distance of each chromosome, which was denoted as the genome-wide pairwise genetic relatedness score $S_{ij}$. We removed the least number of individuals to ensure that $S_{ij}$ between any individual $i$ and $j$ is smaller than 0.24 (i.e. relative removal excluding cousins and grandparents, whilst retaining some rare populations with high inbreeding coefficients). The final UK Biobank dataset has 462,694 individuals (including both males and females).

The local ancestry inference implemented by SparsePainter [17] requires the input of parameter 'fixlambda', i.e. the recombination scaling constant. We estimated it as 128.6 using the Viterbi algorithm on chromosome 20 and used this fixed value for the LAI across all the chromosomes. We set the parameters of SparsePainter to find the 20 longest matches (longer than 10 SNPs) at each SNP, and we used the default setting for all the other parameters. SparsePainter paints in batches which we set to 10,000 individuals to trade-off parallelism with memory storage and data input.

## Simulation for ancestry-specific GWAS

With local ancestry inferred, we are interested in estimating the ancestry-specific effect sizes for GWAS. With accurate and certain local ancestry calls, Tractor [21] has been shown to accurately estimate ancestry-specific effect sizes. At a given SNP, let $Y$ denote the phenotype, $T_k$ denote the number of copies of the risk allele from ancestry $k$, and $X_k$ denote the number of copies at this locus (i.e. either the risk allele or the alternative allele) from ancestry $k$. Also let $C_j (j = 1, 2, ..q)$ denote $q$ different covariates, and $\epsilon$ denote the random error. Under a two-way admixture, the Tractor model for continuous outcome is:

$$Y = \beta_0 + \beta_1 T_1 + \beta_2 T_2 + \beta_3 X_1 + \sum_{j=1}^{q} \beta_{j+3} C_j + \epsilon \tag{1}$$

However, in practice, local ancestry is not known and must be inferred. Many modern local ancestry inference tools, such as SparsePainter [17], ChromoPainter [3], RFMix [27] and FLARE [25] implement Li and Stephens hidden Markov model [60] and therefore can report local ancestry probabilities at each locus. Other software, including FLARE, report ancestry calls as a best guess of local ancestry probabilities, omitting loci that were admixed longer ago and hence are uncertain. Therefore, we performed a simulation study to compare the performance of the Tractor model under different LAI uncertainties, with $T_1$, $T_2$ and $X_1$ computed from either (a) The true local ancestry call (hereafter 'TRUTH'); (b) The raw local ancestry probabilities (hereafter 'RAW'); (c) The best-guess local ancestry call from raw local ancestry probabilities (hereafter 'BESTGUESS'); or (d) Random sampling of local ancestry from raw local ancestry probabilities (hereafter 'SAMPLING').

Below we describe our simulation procedures. The sample size for the simulation is $N = 20,000$ diploid individuals. For simplicity, we simulated two ancestries and therefore considered only the expected value of Ancestry 1 that $a = \{0.1, 0.25, 0.5\}$. To generate individual variation in certainty with a controlled mean that can be interpreted as an 'error rate', we first sampled a latent variable $\tilde{A}_{ikd}$ for the $d$th copy of the $i$th individual for the $k$th ancestry, from Bernoulli distribution: $\tilde{A}_{i1d} \sim \text{Bernoulli}(a)$ and $\tilde{A}_{i2d} = 1 - \tilde{A}_{i1d}$. Since the haplotypes are not observed perfectly, we considered different average confidence, i.e. the probability of LAI: $p = \{0.5, 0.6, 0.7, 0.8, 0.9, 0.95, 0.98\}$ which are assumed to be correctly calibrated. We sampled the haplotype certainties $p_{id}$ for the $d$th copy of individual $i$ from Beta distribution with mean

$E(p_{id}) = p$ and variance $Var(p_{id}) = (0.05/p)^2$. This forms the 'observed' local ancestry probability $p_{ikd} = p_{id}$ if $\tilde{A}_{ikd} = 1$, and $p_{ikd} = 1 - p_{id}$ otherwise. Finally, we sampled the true ancestry call $A^*_{ikd}$ from Bernoulli distribution: $A^*_{i1d} \sim \text{Bernoulli}(p_{i1d})$ and $A^*_{i2d} = 1 - A^*_{i1d}$.

We simulated the genotype $G_{id}$ for the $d$th copy of individual $i$ using the Balding-Nichols model [61] with a fixation index of $F_{st} = 0.2$, which accounts for varying local ancestry $A_{ikd}$. We considered SNPs falling into 4 different MAF thresholds: $1\% \leq f \leq 5\%$, $5\% \leq f \leq 10\%$, $10\% \leq f \leq 20\%$, and $f \geq 20\%$. The ancestral frequency $f$ is Uniform under each MAF threshold, from which the population-specific risk allele frequency for the $k$th ancestry was simulated as $f_k \sim Beta\left(f\frac{(1-F_{st})}{F_{st}}, (1-f)\frac{(1-F_{st})}{F_{st}}\right)$ also with restrictions of the corresponding MAF threshold. The alleles $G_{id}$ were simulated from Bernoulli distribution with their population's frequency: $G_{id} \sim \text{Bernoulli}(f_k)$, where $k = 1$ if $A^*_{i1d} = 0$, and $k = 2$ if $A^*_{i1d} = 1$. $G_{id} = 1$ and $G_{id} = 0$ represent the risk and alternative allele, respectively.

Subsequently, we simulated a trait from the Tractor model using the true haplotype ancestries, i.e. $T_{i1} = G_{i1}A^*_{i11} + G_{i2}A^*_{i12}$ and $T_{i2} = G_{i1}A^*_{i21} + G_{i2}A^*_{i22}$ as the true ancestral dosages, and then generated the phenotype $Y_i = 2 + 5T_{i1} + T_{i2} + \epsilon_i$, where $\epsilon_i \sim N(0, 15^2)$. We used a high variance of the random error because the heritability of a single SNP in practice is always very low.

We then constructed the genetic effect values under the same model but using the *observed* ancestry under different treatments of uncertainty, i.e. $T^S_{i1} = G_{i1}A^S_{i11} + G_{i2}A^S_{i12}$, $T^S_{i2} = G_{i1}A^S_{i21} + G_{i2}A^S_{i22}$ and $X^S_{i1} = A^S_{i11} + A^S_{i12}$, where $S = \{\text{TRUTH}, \text{RAW}, \text{BESTGUESS}, \text{SAMPLING}\}$. Intuitively, $A^{\text{TRUTH}}_{ikd} = A^*_{ikd}$, and $A^{\text{RAW}}_{ikd} = p_{ikd}$. For BESTGUESS, we let $A^{\text{BESTGUESS}}_{ikd} = 1$ if $p_{ikd} \geq 0.5$, and $A^{\text{BESTGUESS}}_{ikd} = 0$ otherwise. For SAMPLING, we sampled from Bernoulli distribution : $A^{\text{SAMPLING}}_{i1d} \sim \text{Bernoulli}(p_{i1d})$ and $A^{\text{SAMPLING}}_{i2d} = 1 - A^{\text{SAMPLING}}_{i1d}$.

Finally, we fit the linear regression model $Y_i \sim \beta^S_0 + \beta^S_1 T^S_{i1} + \beta^S_2 T^S_{i2} + \beta^S_3 X^S_{i1}$. For each scenario, we simulated 1,000 traits and compared the average estimated $\beta^S_1$ and $\beta^S_2$ with the truth, i.e. $\beta_1 = 5$ and $\beta_2 = 1$.

## Computing the distribution of haplotypes across ancestral probabilities by continent

For the subsequent analysis, we focused on 35 phenotypes (the majority of them are diseases) which were reported (in the UK Biobank database) to have at least 30 SNPs significant at $p < 5 \times 10^{-6}$ that overlap our SNP set before and at least 5 after LD pruning. We painted the UK Biobank using the pre-described reference panel, and summarized the painting from 93-population level into 8-continental level, i.e. Europe (all Europe excluding East Europe and UK), East Europe, UK, Africa, South Asia, Middle East, East Asia, and America (see S3 Table for details). Then we aimed to assess the distribution of haplotypes across ancestral probabilities by continent. For each phenotype, we selected the most significant SNP, and haplotype counts (with a total of 925,388, i.e. twice the number of individuals) exceeding specific ancestral probability thresholds ranging from 0.5 to 1.0 (with a step size of 0.01) were computed for all continents (S8 Fig). Median counts and 95% confidence intervals were calculated to summarize the distribution of haplotype numbers at each threshold.

## Computation of ARS

We computed the ARS [35,36] for 35 phenotypes at the 8 continental levels described above. We first filtered all the SNPs which are significant at $p < 10^{-6}$ from the published GWAS results and are contained in our SNP set, and then did LD pruning in PLINK 2.0 [51] based on $R^2$ at a threshold of 0.5.

Let $A_{ijk}$ denote the local ancestry probability of ancestry $k$ for the $i$th haplotype at the $j$th SNP, and $G_{ij}$ denote the genotype of the $i$th haplotype at the $j$th SNP, which takes value 1 or 0 representing the risk or the alternative allele, respectively. Then the risk allele frequency of SNP $j$ for ancestry $k$, $f_{jk}$, is computed as:

$$f_{jk} = \frac{\sum_{i=1}^{2N} A_{ijk}G_{ij}}{\sum_{i=1}^{2N} A_{ijk}} \tag{2}$$

where $N$ is the total number of individuals.

We fit linear and logistic regression models for continuous and binary phenotypes, using sex, age and the top 150 HCs as covariates, and computed the effect size $\beta_j$ for the risk allele of the $j$th SNP. To compute ARS for ancestry $k$, we summed over all $M$ pruned significant SNPs in an additive model:

$$ARS_k = \sum_{j=1}^{M} f_{jk}\beta_j \tag{3}$$

## Uncertainty for ARS

There are two types of randomness involved in the calculation of the ARS. The first is 'randomness over which individuals were in our dataset'. Calculating ARS involves estimating the frequency of a trait-associated SNP in each of the populations, some of which have few samples. The second is 'randomness over the ancestry assigned to trait-associated SNPs'. As relatively few SNPs are sampled for some traits, we might expect a similar random set of SNPs to contain spurious trait associations.

**Randomness in ARS over individuals.**

To obtain a 95% confidence interval for an ARS accounting for the random sample of individuals, we bootstrapped over individuals for 1,000 times. We obtained the 2.5% and 97.5% quantile of risk allele frequency of SNP $j$ for ancestry $k$, $f_{jk}^{lower}$ and $f_{jk}^{upper}$, which replace $f_{jk}$ in Eq (3) to obtain the lower and upper bound of ARS for ancestry $k$, $ARS_k^{lower}$ and $ARS_k^{upper}$.

**Randomness in ARS-ancestry associations.**

To account for variation in ancestry assigned to trait-associated SNPs, we simulate pseudo-ARS under the null that trait-associated SNPs are not associated with ancestry. The pseudo-ARS are generated by matching each associated SNP with a set of LD-pruned SNPs with the same risk allele frequency (precision up to 1%) in the UK Biobank.

To illustrate the variation in SNP ancestries genome-wide, we visually compare the value of the ancestry of each SNP with its pseudo-replicates. S12 Fig shows this per-SNP for the 8 pruned significant SNPs for pulmonary embolism. Some visually extreme ancestral frequency estimates are expected under the null, whilst many significant SNPs for pulmonary embolism are outliers of allele frequencies in specific populations. Whilst this large $F_{st}$ may indicate selection, it only translates to a signal for the target trait if it replicates at other SNPs.

## Assessing the significance level of ARS

To assess whether specific ARS are significant, we combine the above two causes of variation. For each pruned significant SNP $j$ of each phenotype, we found all the SNPs across all 22 chromosomes with the same allele frequency (precision up to 1%), and used LD pruning

to remove chance duplications. Then we bootstrapped over SNPs, i.e. resampled with replacement, for 50,000 times, and obtained the simulated risk allele frequency $f_{ijk}^*$ for the $k$th population, where $i = 1, \dots, 50000$ denotes the index of the bootstrapped SNPs. Next, we used the real risk-allele effect size $\beta_j$ to compute the simulated ARS: $ARS_{ik}^* = \sum_{j=1}^M f_{ijk}^* \beta_j$.

We then designed a test for whether ARS for the $k$th population is significantly different from the population mean under the null that the variation observed is driven by uncertainty in frequency estimates (bootstrap over individuals) and randomness in which SNPs are associated with ancestry. For this, we sampled a realization of the ARS accounting for uncertainty: $\widetilde{ARS}_{ik} \sim N(ARS_{ik}^*, \sigma(ARS_k)^2)$, where $\sigma(ARS_k)$ is the standard error obtained by bootstrap over individuals. From this, we can compute an empirical test statistic accounting for randomness in SNP ancestry calls. We computed a two-sided p-value for each population as two times the proportion that (observed, individual-variation accounted) $\widetilde{ARS}_{ik}$ is more extreme than (ancestry sampling of SNP accounted) $ARS_k$ (S14 Fig). We then used Benjamini-Hochberg correction [39] on the p-values to control the false discovery rate.

We finally used Fisher's method [62] to combine the p-values $p_k$ of all $K = 8$ populations as an aggregated p-value for each phenotype. In detail, we computed the test statistic $\chi_{2K}^2 = -2\sum_{k=1}^K \ln(p_k)$, and then computed the p-value for $\chi_{2K}^2$ with $2K$ degrees of freedom. Finally, the Benjamini-Hochberg corrected p-values are reported in S13 Fig.

## Converting educational attainment from categorical variable to continuous variable

Educational attainment is converted into a 'years of education' score using the International Standard Classification for Education (ISCED) definition [63]:

$$ES = \begin{cases} 7, & \text{if none of the above} \\ 10, & \text{if CSEs or equivalent or O levels/GCSEs or equivalent} \\ 13, & \text{A levels/AS levels or equivalent} \\ 15, & \text{Other professional qualifications eg: nursing, teaching} \\ 19, & NVQ \text{ or HND or HNC or equivalent} \\ 20, & \text{College or University degree} \end{cases}$$

## Supporting information

**S1 Table. Average out-of-sample $R^2$ explained by the top 150 HCs and top 18 PCs and the optimal number of top HCs and PCs for continuous phenotypes.**
(XLSX)

**S2 Table. Average out-of-sample Mcfadden's $R^2$ explained by the top 150 HCs and top 18 PCs and the optimal number of top HCs and PCs for binary phenotypes.**
(XLSX)

**S3 Table. 93 worldwide ancestries and their abbreviations as summarized from public-available reference datasets.**
(XLSX)

**S4 Table. Distribution of n = 20,092 individuals by birthplace and self-reported ethnic group in the study assessing the performance of HCs and PCs to predict birthplaces from the UK Biobank.**
(XLSX)

**S1 Fig. Confusion matrices for the performance of HCs and PCs on predicting worldwide birthplaces visualized in heatmaps.**
(TIFF)

**S2 Fig Average out-of-sample $R^2$ explained by different numbers of top HCs/PCs for 24 continuous phenotypes.** This study includes n = 406,773 UK Biobank individuals.
(TIFF)

**S3 Fig Average out-of-sample $R^2$ explained by different numbers of top HCs/PCs for 53 binary phenotypes.** The points and lines are faded for negative average out-of-sample $R^2$. This study includes n = 406,773 UK Biobank individuals.
(TIFF)

**S4 Fig Comparison between HC-corrected and PC-corrected GWAS for 26 continuous phenotypes.** Each point represents its -log10(P-value) from GWAS corrected by the top 18 HCs or PCs. This study includes n = 406,773 UK Biobank individuals.
(TIFF)

**S5 Fig Comparison between HC-corrected and PC-corrected GWAS for 53 binary phenotypes.** Each point represents its -log10(P-value) from GWAS corrected by the top 18 HCs or PCs. This study includes n = 406,773 UK Biobank individuals.
(TIFF)

**S6 Fig Comparison of the average estimated effect size of ancestries from the Tractor model with different methods to represent local ancestries under $\mathbf{E(A_{i1d}^*)} = \mathbf{0.25}$.** $E(A_{i1d}^*)$ denotes the average probability of Ancestry 1. The x-axis represents the average certainty of LAI (see Methods) and the y-axis represents the average estimated effect sizes for Ancestry 1 (plot a) and Ancestry 2 (plot b). Different MAF thresholds $f$ are compared: $1\% \le f \le 5\%$, $5\% \le f \le 10\%$, $10\% \le f \le 20\%$, and $f \ge 20\%$. The simulation was repeated 1,000 times with n = 20,000 diploid individuals.
(TIFF)

**S7 Fig Comparison of the average estimated effect size of ancestries from the Tractor model with different methods to represent local ancestries under $\mathbf{E(A_{i1d}^*)} = \mathbf{0.5}$.** $E(A_{i1d}^*)$ denotes the average probability of Ancestry 1. The x-axis represents the average certainty of LAI (see Methods) and the y-axis represents the average estimated effect sizes for Ancestry 1 (plot a) and Ancestry 2 (plot b). Different MAF thresholds $f$ are compared: $1\% \le f \le 5\%$, $5\% \le f \le 10\%$, $10\% \le f \le 20\%$, and $f \ge 20\%$. The simulation was repeated 1,000 times with n = 20,000 diploid individuals.
(TIFF)

**S8 Fig Distribution of LAI inferred with SparsePainter in UK Biobank samples across ancestral probabilities by continent**, using 1000 Genomes as reference. The x-axis represents the local ancestry probabilities, and the y-axis represents the number of haplotypes measured as two times the number of individuals painted in the UK Biobank on a log10 scale. There are a total of n = 925,388 haplotypes. Each plot displays the median cumulative haplotype count (blue dashed line) along with the corresponding 95% confidence intervals (shaded regions) at varying local ancestry probability thresholds, based on all pruned significant SNPs across 35 phenotypes (investigated in the ARS analysis) for 8 ancestries. Median values at the 98% (red dashed) and 95% (green dashed) probability thresholds are annotated.
(TIFF)

**S9 Fig Comparison of the standard error of the estimated effect size of ancestries from the Tractor model with different methods to represent local ancestries under $E(A^*_{i1d})$ = 0.1.** $E(A^*_{i1d})$ denotes the average probability of Ancestry 1. The x-axis represents the average certainty of LAI (see Methods) and the y-axis represents the average standard error of the estimated effect size of Ancestry 1 (plot a) and Ancestry 2 (plot b). Different MAF thresholds $f$ are compared: $1\% \le f \le 5\%$, $5\% \le f \le 10\%$, $10\% \le f \le 20\%$, and $f \ge 20\%$. The simulation was repeated 1,000 times with n = 20,000 diploid individuals.
(TIFF)

**S10 Fig Comparison of the standard error of the estimated effect size of ancestries from the Tractor model with different methods to represent local ancestries under $E(A^*_{i1d})$ = 0.25.** $E(A^*_{i1d})$ denotes the average probability of Ancestry 1. The x-axis represents the average certainty of LAI (see Methods) and the y-axis represents the average standard error of the estimated effect size of Ancestry 1 (plot a) and Ancestry 2 (plot b). Different MAF thresholds $f$ are compared: $1\% \le f \le 5\%$, $5\% \le f \le 10\%$, $10\% \le f \le 20\%$, and $f \ge 20\%$. The simulation was repeated 1,000 times with n = 20,000 diploid individuals.
(TIFF)

**S11 Fig Comparison of the standard error of the estimated effect size of ancestries from the Tractor model with different methods to represent local ancestries under $E(A^*_{i1d})$ = 0.5.** $E(A^*_{i1d})$ denotes the average probability of Ancestry 1. The x-axis represents the average certainty of LAI (see Methods) and the y-axis represents the average standard error of the estimated effect size of Ancestry 1 (plot a) and Ancestry 2 (plot b). Different MAF thresholds $f$ are compared: $1\% \le f \le 5\%$, $5\% \le f \le 10\%$, $10\% \le f \le 20\%$, and $f \ge 20\%$. The simulation was repeated 1,000 times with n = 20,000 diploid individuals.
(TIFF)

**S12 Fig Simulated distribution of ancestral allele frequency for pulmonary embolism.** The error bars represent the 95% confidence interval of ancestral allele frequency. The distribution of simulated allele frequency for each population of all the matched SNPs in the UK Biobank is shown as a raincloud plot. The dashed black lines represent the genome-wide allele frequency.
(TIFF)

**S13 Fig Phenotype-wise association of ARS across populations using Fisher's method.** The y-axis shows the -log10 scale of the p-values derived from the ARS calculated across different populations and then aggregated using Fisher's method, i.e. chi-squared test (see Methods).
(TIFF)

**S14 Fig ARS for 35 phenotypes ordered by the overall significance level.** ARS are computed from n = 462,694 individuals in the UK Biobank. The error bars represent the 95% confidence interval of ARS. The distribution of simulated ARS for each population is shown as a raincloud plot. The dashed black lines represent the average ARS of all populations weighted by the population sizes. The ARS of populations with raw p-values significant at 5% computed from the two-sided empirical test are annotated.
(TIFF)

## Acknowledgments

This work was carried out using the computational facilities of the Advanced Computing Research Centre, University of Bristol - http://www.bris.ac.uk/acrc.

## Author contributions

**Conceptualization:** Yaoling Yang, Daniel J. Lawson.

**Data curation:** Yaoling Yang, Daniel J. Lawson.

**Formal analysis:** Yaoling Yang, Daniel J. Lawson.

**Funding acquisition:** Yaoling Yang, Daniel J. Lawson.

**Investigation:** Yaoling Yang, Daniel J. Lawson.

**Methodology:** Yaoling Yang, Daniel J. Lawson.

**Project administration:** Yaoling Yang, Daniel J. Lawson.

**Resources:** Yaoling Yang, Daniel J. Lawson.

**Software:** Yaoling Yang.

**Supervision:** Daniel J. Lawson.

**Validation:** Yaoling Yang, Daniel J. Lawson.

**Visualization:** Yaoling Yang, Daniel J. Lawson.

**Writing – original draft:** Yaoling Yang, Daniel J. Lawson.

**Writing – review & editing:** Yaoling Yang, Daniel J. Lawson.

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
