## [Decision Letter · Decision Letter 0]

9 Jul 2025

PGENETICS-D-25-00322

From individuals to ancestries: towards attributing trait variation to haplotypes

PLOS Genetics

Dear Dr. Lawson,

Thank you for submitting your manuscript to PLOS Genetics and we apologize for the delay in making a decision. Overall, the reviewers were generally positive about your work but raised several issues that need to be addressed. Therefore, we invite you to submit a revised version of the manuscript that addresses the points raised during the review process.

Please submit your revised manuscript within 60 days Sep 07 2025 11:59PM. If you will need more time than this to complete your revisions, please reply to this message or contact the journal office at plosgenetics@plos.org. Please include the following items when submitting your revised manuscript:

We look forward to receiving your revised manuscript.

Kind regards,

Elizabeth G Atkinson, Ph.D.

Guest Editor

PLOS Genetics

Michael Epstein

Section Editor

PLOS Genetics

Aimée Dudley

Editor-in-Chief

PLOS Genetics

Anne Goriely

Editor-in-Chief

PLOS Genetics

**Journal Requirements:**

At this stage, the following Authors/Authors require contributions: Yaoling Yang, and Daniel Lawson. Please ensure that the full contributions of each author are acknowledged in the "Add/Edit/Remove Authors" section of our submission form.

The list of CRediT author contributions may be found here: https://journals.plos.org/plosgenetics/s/authorship#loc-author-contributions

Potential Copyright Issues:

- Figures 1A and 1B. Please (a) provide a direct link to the base layer of the map (i.e., the country or region border shape) and ensure this is also included in the figure legend; and (b) provide a link to the terms of use / license information for the base layer image or shapefile. We cannot publish proprietary or copyrighted maps (e.g. Google Maps, Mapquest) and the terms of use for your map base layer must be compatible with our CC BY 4.0 license.

**Reviewers' comments:**

Reviewer's Responses to Questions

**Comments to the Authors:**

Reviewer #1: This manuscript explores the utility of haplotype-based representations, specifically haplotype components (HCs), for improving population structure inference and trait association studies in human genetics. Leveraging UK Biobank data, the authors compare HCs with traditional principal components (PCs) in a variety of contexts: birthplace prediction, phenotype prediction, GWAS correction, ancestry-specific effect size estimation (via Tractor), and the development of Ancestral Risk Scores (ARS). The authors argue that HCs provide more nuanced representations of population structure and that ARS, rather than ancestry-specific effect estimates, may be more robust in admixed populations. Overall, this manuscript is well-written. The proposed simulation studies and real data analyses support their conclusion. I have a few comments as follows.

1. The analyses focus almost exclusively on the UK Biobank, where the vast majority of participants are of British ancestry. The authors could discuss how the performance of HCs may vary across more admixed or globally diverse datasets (e.g., All of Us or PAGE). Additional simulations mimicking more complex admixture scenarios would strengthen this claim, while I understand it could be of future work.

2. While HCs outperform PCs in various predictive tasks, their biological interpretation remains unclear. For example, the authors mentioned that the optimal number of HCs is as large as 150 but for PCs is only 18. More explanation on interpreting the selected HCs would improve the generalizability of the results.

3. The ARS approach is of potential interest, but the real-world utility is somewhat speculative, especially given that associated SNPs were primarily derived from European-centric GWAS. The authors should discuss more explicitly the implications of using ARS constructed from European-discovered variants when applied to other populations.

4. The paper underscores the difficulty of accurate LAI and its impact on Tractor and related frameworks. It could be helpful for the authors to discuss how probabilistic LAI models (e.g. FLARE) can be leveraged for ancestry-specific effect size estimation.

Other comments:

1. Line 139, should it be “every PC > 18”?

2. Line 334, “PC” should be “PCs”?

3. Line 343, should it be “The top 150 HCs and 18 PCs”?

Reviewer #2: Overall, I thought this was a timely, important and interesting paper, building on recent results published this year on the importance of haplotype-based methods for GWAS. The authors firstly confirm previous results that haplotype components are better at capturing population stratification than normal PCA analyses, and show that HCs have greater predictive power for a variety of traits because of this. However, they do show that despite this GWAS results for UKBB do not change much, which they hypothesise is because of the low amounts of structure. The completely novel results of the paper are in the second section on local ancestry inference and association studies, where through simulations the authors show that ancestry-specific GWAS is poor when LAI confidence is low. They present and discuss the use of Ancestral Risk Scores as an alternative for understanding population-specific risk.

The paper provides multiple novel results and statistics, building on methods presented in Yang et al., 2025. It opens up avenues of research using datasets with more complex ancestry patterns than UKBB, and I think will be of interest to the GWAS/ population genetics community. I have some minor comments to improve the clarity of the manuscript.

Abstract- I thought the only new GWAS hits were for education attainment- ‘birthplace and lifestyle related phenotypes’ seem misleading?

Line 47- check this ref, I think they are saying 40% of variation in traits is dues to SNPs, not 40% of associations are between SNPs and traits.

Fig 1- could you include some discussion on the birthplaces with very low predictive power- is it a sample size thing, are they always classified as a nearby country? Similarly there are a few cases where PCs seems to do better, e.g. Bulgaria and Croatia- any ideas on this, maybe they are cases with lots of recent migration between countries throwing off HCs? Overall caption of Fig 1 could use more detail on the analysis- 1d looks a lot like a PCA without reading all the text.

Line 164- state how many HCs and PCs did you use for correction here- this information is only in supplementary. Also did you try the GWAS with >18 HCs, and see any changes in result?

Line 170- examples of other phenotypes?

Line 180- this section could use just a few more sentences describing what you did to improve clarity- a little more about your simulations, and an explanation of truth, raw etc would be useful (and why you picked these).

Additionally the results from supplementary fig. 8 seem very important for explaining the context of your simulation result but are poorly explained- I think you used sparsepainter for the probabilities but this isn’t explicitly mentioned anywhere? The total number of haplotypes that exceed >0.5 probability is different for each continent depending on the sample, so could you report this alongside the median? Maybe some (very brief) background on different LAI methods and their accuracies could be useful here. Overall I’m still a bit confused- it seems TRACTOR approaches do have merit in datasets of admixed inds with ancestry from multiple continents (where LAI should be relatively accurate), but UKBB does not have many of these individuals (or of non-UK individuals in general) AND LAI probs are lower, so will have low power for ancestry specific effects?

Line 211- Again, just another sentence or two on ARS and how they are calculated would be useful at the start of this section for context. The A_ijk used in the ARS calculation are the same as those shown in Fig s8?

Did you try ARS on your smaller reference groupings, rather than just the continental ones?

Line 299- was test and training set randomised at each repeat?

**Have all data underlying the figures and results presented in the manuscript been provided?**

Reviewer #1: Yes

Reviewer #2: Yes

PLOS authors have the option to publish the peer review history of their article (what does this mean?). If published, this will include your full peer review and any attached files.

Reviewer #1: No

Reviewer #2: No

**Figure resubmission:**
---

## [Decision Letter · Decision Letter 1]

15 Sep 2025

Dear Dr Lawson,

We are pleased to inform you that your manuscript entitled "From individuals to ancestries: towards attributing trait variation to haplotypes" has been editorially accepted for publication in PLOS Genetics. Congratulations!

In the meantime, please log into Editorial Manager at https://www.editorialmanager.com/pgenetics/ click the "Update My Information" link at the top of the page, and update your user information to ensure an efficient production and billing process. Note that PLOS requires an ORCID iD for all corresponding authors. Therefore, please ensure that you have an ORCID iD and that it is validated in Editorial Manager. To do this, go to ‘Update my Information’ (in the upper left-hand corner of the main menu), and click on the Fetch/Validate link next to the ORCID field.  This will take you to the ORCID site and allow you to create a new iD or authenticate a pre-existing iD in Editorial Manager.

Yours sincerely,

Elizabeth G Atkinson, Ph.D.

Guest Editor

PLOS Genetics

Michael Epstein

Section Editor

PLOS Genetics

Aimée Dudley

Editor-in-Chief

PLOS Genetics

Anne Goriely

Editor-in-Chief

PLOS Genetics

Comments from the reviewers (if applicable):

Reviewer #1:

Reviewer #2:

Reviewer's Responses to Questions

**Comments to the Authors:**

Reviewer #1: I thank the authors in addressing my previous comments, which have further enhanced the clarity of the manuscript. I do not have further questions and would like to recommend this manuscript for publication.

Reviewer #2: All my concerns have been addressed and the clarity of the manuscript is much improved. I am happy for it to be published in its current form.

**Have all data underlying the figures and results presented in the manuscript been provided?**

Reviewer #1: Yes

Reviewer #2: Yes

PLOS authors have the option to publish the peer review history of their article (what does this mean?). If published, this will include your full peer review and any attached files.

Reviewer #1: **Yes: **Xihao Li

Reviewer #2: No

**Data Deposition**

http://datadryad.org/submit?journalID=pgenetics&manu=PGENETICS-D-25-00322R1

**Press Queries**

---

## [Editor Report · Acceptance letter]

PGENETICS-D-25-00322R1

From individuals to ancestries: towards attributing trait variation to haplotypes

Dear Dr Lawson,

We are pleased to inform you that your manuscript entitled "From individuals to ancestries: towards attributing trait variation to haplotypes" has been formally accepted for publication in PLOS Genetics! Your manuscript is now with our production department and you will be notified of the publication date in due course.

With kind regards,

Anita Estes

PLOS Genetics

On behalf of:
